# Bones or Stones: How Can We Apply Geophysical Techniques in Bone Research?

**DOI:** 10.3390/ijms251910733

**Published:** 2024-10-05

**Authors:** Zoltán Szekanecz, Anikó Besnyi, Péter Kónya, Judit Füri, Edit Király, Éva Bertalan, György Falus, Beatrix Udvardi, Viktória Kovács-Kis, László Andrássy, Gyula Maros, Tamás Fancsik, Zsófia Pethő, Izabella Gomez, Ágnes Horváth, Katalin Gulyás, Balázs Juhász, Katalin Hodosi, Zsuzsanna Sándor, Harjit P. Bhattoa, István J. Kovács

**Affiliations:** 1Department of Rheumatology, Faculty of Medicine, University of Debrecen, 4032 Debrecen, Hungary; pethozsofi0331@gmail.com (Z.P.); gomez.izabella@gmail.com (I.G.); kis.horvathagi@gmail.com (Á.H.); gulyaskata@yahoo.com (K.G.); khodosi@gmail.com (K.H.); 2Supervisory Authority for Regulatory Affairs, Geological Survey, 1143 Budapest, Hungary; aniko.besnyi@sztfh.hu (A.B.); peter.konya@sztfh.hu (P.K.); judit.furi@sztfh.hu (J.F.); edit.kiraly0305@gmail.com (E.K.); bertalan1955@gmail.com (É.B.); gyorgy.falus@sztfh.hu (G.F.); idosandrassy@gmail.com (L.A.); gyula.maros1@gmail.com (G.M.); tamas.fancsik@sztfh.hu (T.F.); 3Kentucky Geological Survey, University of Kentucky, Lexington, KY 40506, USA; 4TÜV Rheinland InterCert Ltd., 1143 Budapest, Hungary; udvbeatrix@gmail.com; 5HUN-REN Centre for Energy Research, 1121 Budapest, Hungary; kis.viktoria@ek.hun-ren.hu; 6National Institute of Rheumatology and Physiotherapy, 1023 Budapest, Hungary; 7Department of Oncology, Faculty of Medicine, University of Debrecen, 4032 Debrecen, Hungary; juhaszbalazs@hotmail.com; 8National Institute of Oncology, 1122 Budapest, Hungary; saniel11@freemail.hu; 9Department of Laboratory Medicine, Faculty of Medicine, University of Debrecen, 4032 Debrecen, Hungary; harjit@med.unideb.hu; 10HUN-REN Institute of Earth Physics and Space Science, 1052 Budapest, Hungary; kovacs.istvan.janos@epss.hun-ren.hu

**Keywords:** bone, X-ray diffraction, thermogravimetry, Fourier-transform infrared spectrometry with attenuated total reflectance accessory, inductively coupled mass spectrometry, inductively coupled optical emission spectrometry

## Abstract

Some studies have used physical techniques for the assessment of bone structure and composition. However, very few studies applied multiple techniques, such as those described below, at the same time. The aim of our study was to determine the chemical and mineralogical/organic composition of bovine tibial bone samples using geophysical/geochemical reference techniques. X-ray diffraction (XRD), thermogravimetry (TG), Fourier-transform infrared spectrometry with attenuated total reflectance accessory (FTIR-ATR), inductively coupled mass spectrometry (ICP-MS) and inductively coupled optical emission spectrometry (ICP-OES) were applied to measure the organic and inorganic composition of 14 bovine bone samples. In addition, peripheral quantitative CT (QCT) was used to assess BMD in these bones. We were able to define the total composition of the studied bone samples. ICP-OES and ICP-MS techniques were used to determine the major and trace element composition. The X-ray analysis could detect inorganic crystalline compounds of bones, such as bioapatite, and its degree of ordering, indicating whether the bones belong to a younger or older individual. The total volatile content of the samples was calculated using TG and resulted in about 35 weight% (wt%). This, together with the 65 wt% total resulting from the chemical analysis (i.e., inorganic components), yielded a total approaching 100 wt%. As a large portion of the volatile content (H_2_O, CO_2_, etc.) was liberated from the organic components and, subordinately, from bioapatite, it could be concluded that the volatile-to-solid ratio of the examined bone samples was ~35:65. The FTIR-ATR analysis revealed that the organic portion consists of collagens containing amide groups, as their typical bands (OH, CH, CO, NC) were clearly identified in the infrared spectra. Numerous parameters of bone composition correlated with BMD as determined by QCT. In conclusion, we performed a complex evaluation of bovine bones to test multiple geophysical/geochemical techniques in bone research in association with QCT bone densitometry. From a medical point of view, the composition of the studied bones could be reliably examined by these methods.

## 1. Introduction

The in vivo assessment of bone structure and bone content can be essential in various bone and calcium homeostatic disorders. Bone densitometry (DEXA) has been widely used to determine bone density in osteoporosis (OP) and other conditions. Bone structure can be assessed by micro-CT, high-resolution MRI techniques or bone biopsy in vivo [1].

These issues lead to the idea of applying geological/geophysical/geochemical techniques to bone research. Indeed, inorganic minerals are the most abundant constituents of bones. Furthermore, as described later, there have been a few initial attempts using geological methods, mostly laser-based, to assess the chemical composition of bones. The main reason for conducting these studies was that we built a great collaboration between rheumatologists and geophysicists/geochemists. As discussed later, we wished to apply a plethora of different techniques to analyze bone structure and composition in great detail.

The determination of the concentration of various minerals, their structure, and their major and trace element content in the bone can have significant relevance within and beyond the field of medicine. The amount of various elements can differ between healthy and unhealthy bones. Furthermore, certain trace elements, such as strontium (Sr), used in the treatment of osteoporosis, lithium (Li), used to treat psychiatric and thyroid diseases, and aluminum (Al) and lanthanum (La), used in hemodialyzed patients, can accumulate in the bone. The determination of these elements could be used in toxicology studies. Forensic medicine could use such techniques to assess the accumulation of lead (Pb) or arsenic (As) in various tissues [2,3,4,5]. Additionally, laser techniques have also been used to assess bone age in archeology [4,5]. Nonetheless, to date, very few studies, mostly carried out on animals using inductively coupled plasma spectrometry (ICP-MS) and/or inductively coupled plasma optical emission spectrometry (ICP-OES), have been published [2,3,4,5,6,7]. Certainly, other types of spectroscopy, such as Raman [8,9], energy-dispersing X-ray (EDX) [10,11], atomic absorption spectroscopy [12,13], X-ray fluorescence [14,15], and others, have also been applied to bone and osteoporosis research. Yet, we used ICP-MS and ICP-OES as the available techniques suitable for complex bone research.

Many elements can play a role in bone physiology and pathology. Al, Cd, Mo, Pb, Zn, Cu, Li, Mg, Mn, Fe, Sr and their salts have been implicated in bone mineralization, osteoblast and osteoclast function, bone stiffness and elasticity and numerous aspects of bone pathology. For example, Al deposition concurrent to hemodialysis or the excessive use of antacids can inhibit bone mineralization and can lead to osteodystrophy or osteomalacia [16]. The phosphate-binding La, also used in nephrology, is a functional mimetic of calcium (Ca). It can accumulate in bone and can exert both physiological and pathological effects [17]. Cd is an antagonist of Ca and vitamin D absorption, and its excess can lead to osteoporosis and osteomalacia with fractures or kidney stones [18]. Mo toxicity leads to impaired bone development and neurodegeneration [19]. Acute Pb toxicity can be determined from blood, but cumulative intoxication can be assessed in bone. Pb inhibits fracture healing [20]. Zn and Cu have important physiological relevance for bone. Zn and Cu deficiency can lead to impaired bone matrix synthesis and osteoporosis [21,22]. Li carbonate has been used to treat depression and thyroid disorders. Li toxicity can lead to hyperparathyroidism and bone loss [23]. Mg is crucial for bone formation and homeostasis. Mg deficiency can lead to increased resorption, impaired elasticity and bone loss [22]. Mn is a natural constituent of bone. Mn toxicity can be observed among miners and factory workers. There have been attempts using a neutron-activation assay to assess Mn content in the bones of the hands [24]. Excess of Fe often leads to bone loss. Fe can inhibit osteoblasts and stimulate osteoclast function [25]. With the introduction of Sr ranelate, an anti-osteoporosis agent, the determination of Sr in the bone may be necessary [26].

Thus, bone research and geology/geophysics/geochemistry may complement each other, and this may open a new perspective for the development and application of geological techniques to study the chemical and physical nature of bones both within and outside the area of medicine. Recently, we have applied laser-induced plasma spectrometry (LIPS) to bone research. We assessed CaO content and distribution in bovine bone samples and we also compared LIPS to quantitative computed tomography (QCT) [27,28].

We wished to fully characterize the chemical and mineralogical/organic composition of bones. In order to achieve this, a very wide range of geological reference analytical tools, such as X-ray diffraction (XRD) [29,30,31], thermogravimetric analysis (TGA) [31,32,33], Fourier-transform infrared spectrometry with attenuated total reflectance accessory (FTIR-ATR) [34,35,36], inductively coupled optical emission spectrometry (ICP-OES) [27,37,38] and inductively coupled mass spectrometry (ICP-MS) [2,3,4,27,39] combined with laser ablation technique, were applied to measure the organic and inorganic composition of bovine tibia bone samples. These data were compared to a standard technique to determine bone mineral density (BMD) and bone structure, QCT [27,40]. There have been multiple studies applying XRD, TGA and FTIR to bone research starting in the 1950s, especially in the field of bone structure and regenerative medicine (among others: [41,42,43,44,45,46,47,48]). Certainly, as described above, many studies applied these techniques or other forms of spectrometry to bone research [8,9,10,11,12,13,14,15,41,42,43,44,45,46,47,48]. However, most studies used only a few techniques at one time and there have been very few complex studies where all these techniques were employed. Moreover, we did not find any studies where such geophysical techniques were combined with BMD studies using QCT. The application of multiple geological techniques combined with QCT to bone research could yield additional information on bone structure and composition during osteoporosis research.

## 2. Results

### 2.1. Assessment of Bone Density and Physico-Chemical Characteristics of Bovine Bones

The results of these analyses are presented in Table 1. The mean (±SD) age of the 14 animals was 1816.6 ± 1093.6 days (4.97 ± 3.00 years).

Bone densitometry assessment by QCT revealed that the total, trabecular and cortical volumetric BMD of the 14 bovine bones were 954.5 ± 59.3, 647.3 ± 114.9 and 1205.0 ± 22.1 mg/cm^3^, respectively (Table 1). Of course, bovine bone BMD has no “normal values”. Bulk major and trace element composition of the bovine bone samples were determined by ICP-OES and ICP-MS (Table 1). Cu, Li and Zn could be detected by ICP-OES, but As could not (Table 1). Among the elements assessed by ICP-MS, Ba, Ce, Co, Cr, Cu, La, Mn, Mo, Ni, Pb, Rb, Sb, Sn, Sr and Zn were found in detectable amounts in the bones. On the other hand, Bi, Cd, Cs, Nd, Pr, Th, Tl, U and Y could not be detected. The mean concentrations of the detectable elements in the bone samples are shown in Table 1. Then, we moved forward to assess the mineral components of the bone by XRD.

Among the mineral components of the bone, XRD mainly detected apatite. A representative graph of the XRD analysis carried out on five randomly selected bovine bone samples is presented in Figure 1. The mean apatite peak intensity was 1590 ± 70 AU. The ratios of either 211 and 222 or 211 and 300 peak intensity were 5.10 ± 0.31 and 1.77 ± 0.09, respectively. The mean degree of crystallinity was 0.01 ± 0.01%. The mean c-axis was 6.89 ± 0.01 Å (Table 1). However, the exact type of the apatite (OH^−^, F^−^ or Cl^−^ dominance among the anions) could not be unambiguously constrained. Due to the relatively low total inorganic/mineral content of the bones and the fact that the XRD could mainly detect apatite as a mineral component, it seemed justified to carry out complementary TGA and FTIR-ATR assessments to determine the remaining inorganic/mineral and organic phases of the bones.

TGA determines weight loss in parallel with increasing temperature. Upon heating, the colour of the bone samples changed from light grey to whitish yellow. The weight loss of the bone samples could be differentiated into four distinct phases. In the first phase (~45 °C–205 °C), the water, partly bound to the organic material, was lost. In the second phase (~200 °C–400 °C), the organic matter was burnt out during an exothermic reaction. In the third phase (~400 °C–650 °C), the remaining organic material and the OH^-^ content were lost. Finally, in the fourth phase (>650 °C), along with a small decrease in weight, the carbonates were most probably decomposed. An attempt was made to determine the composition of carbonated hydroxyapatite from the amount of CO_3_ lost. However, this was complicated by the presence of poorly crystallized carbonates (i.e., calcite and dolomite) that could have also contributed to the released CO_2_ content [33]. The exothermic band (732 °C–799 °C) suggests a phase transition from apatite to β-tricalcium phosphate. Moreover, the exothermic reaction does not happen in the bone samples treated with additional H_2_O_2_. In the TGA analysis, the total volatile content was 28.56 ± 3.01 wt%. The amount of H_2_O, simple and composite organic content were 6.94 ± 0.66 wt%, 11.03 ± 1.12 wt% and 9.08 ± 1.24 wt%, respectively. The carbonate content was 9.08 ± 1.24 wt% (Table 1). However, the TGA analysis did not get us closer to learning the composition of the organic matter in the bone samples. Therefore, the samples were further investigated by FTIR-ATR.

A representative graph of the FTIR-ATR analysis carried out in five randomly selected bone samples is presented in Figure 2. In the FTIR-ATR analysis the integrated area of H_2_O + CH, CH, PO_4_ + CO_3_ and CO_3_ were 10.07 ± 1.46 cm^−1^, 0.20 ± 0.02 cm^−1^, 8.90 ± 0.95 cm^−1^ and 0.10 ± 0.02 cm^−1^, respectively. The integrated area of amide I + CO_3_ and amide I were 7.65 ± 0.78 cm^−1^ and 1.36 ± 0.18 cm^−1^, respectively. The area of the ratio of amide I to PO_4_ + CO_3_ and that of the CO_3_ to PO_4_ + CO_3_ ratio were 6.61 ± 0.59 and 0.01 ± 0.00, respectively (Table 1).

### 2.2. Correlation between QCT Bone Density and Physico-Chemical Parameters

The results of the Pearson’s correlation analysis are included in Table 2. In the 14 bone samples, total BMD positively correlated with all TGA and FTIR-ATR parameters (*p* < 0.05). Trabecular BMD correlated with most TGA and FTIR-ATR parameters (*p* < 0.005). Cortical BMD exhibited correlations with H_2_O and the total volatile content determined by TGA and with PO_4_ + CO_3_ and CO_3_ analysed by FTIR-ATR (*p* < 0.005). With respect to chemistry, total and trabecular BMD both positively correlated with CaO and negatively with SiO_2_ (*p* < 0.005). Cortical BMD showed positive correlations with CaO, MgO and Na_2_O and negative correlations with Fe_2_O_3_ and SiO_2_ (*p* < 0.005). Regarding ICP-MS, total BMD positively correlated with Co and Sn and negatively with Ni. Total BMD exhibited a positive correlation with Sn and a negative correlation with Ni. Finally, cortical BMD correlated with Co, Mn and Mo and negatively with Ni (*p* < 0.005) (Table 1).

Univariable and multivariable regression analyses were performed to evaluate independent determinants of total, trabecular and cortical BMD (Table 3). In the univariable analysis, the age of animals was an independent determinant of total and trabecular BMD but not of cortical BMD (*p* < 0.005). Most TGA and FTIR-ATR parameters determined total and trabecular BMD. In addition, H_2_O and total volatile content (TGA), as well as PO_4_ + CO_3_ and CO_3_ (FTIR-ATR), also determined cortical BMD (*p* < 0.005). In the chemical analysis, CaO was a positive determinant of total, trabecular and cortical BMD, and SiO_2_ was a negative one. In addition, MgO and Na_2_O also positively determined cortical BMD (*p* < 0.005). Finally, in the ICP-MS analysis, Co positively determined total and cortical BMD, Sn correlated with total and trabecular BMD, Mo determined cortical BMD, while Ni was negatively correlated with total, trabecular and cortical BMD (*p* < 0.005). The multivariable regression analysis confirmed that age can be an independent determinant of total and trabecular BMD, and CaO and MgO can positively determine cortical BMD, while Ni can be negatively correlated with total and trabecular BMD (*p* < 0.005) (Table 3).

### 2.3. Correlations between Physico-Chemical Parameters

Many correlations were found between the chemistry, ICP-OES, ICP-MS, XRD, TGA and FTIR-ATR analyses. These could not be presented in a single large table. Therefore, these correlations are included in Appendix A.

Nine element oxides were included in the chemical analysis. These could be divided into two groups. In general, CaO, MgO, Na_2_O, P_2_O_5_, SO_3_ and SrO positively correlated with most XRD, FTIR-ATR and TA parameters. On the other hand, Fe_2_O_3_, K_2_O and SiO_2_ exhibited rather negative correlations with these markers (*p* < 0.005) (Appendix A).

In the ICP-OES analysis, Cu showed several correlations, while Li and Zn showed only a few correlations. Cu positively correlated with the degree of crystallinity (XRD), with most TGA and FTIR-ATR parameters, and with Fe_2_O_3_, K_2_O, SiO_2_ (chemical analysis), La, Rb and Sb (ICP). Cu inversely correlated with Co and Sn as determined by ICP-MS (*p* < 0.005) (Appendix A).

Altogether, fifteen elements could be detected in the bone samples by ICP-MS. Among them, Ba, Co, La, Ni, Sb, Sn and Sr showed a great number of correlations with XRD, FTIR-ATR and TGA parameters. These elements could be divided into two groups. In general, Ba, Co, Sn and Sr showed positive correlations with most XRD, TGA and FTIR-ATR parameters, while La, Ni and Sb were inversely correlated with these markers (*p* < 0.005). Ce, Cr, Cu, Mn, Mo, Pb, Rb and Zn showed very few correlations. Therefore, their correlation patterns with XRD, TGA and FTIR-ATR parameters could not be unequivocally determined (Appendix A).

In general, XRD parameters negatively correlated with simple organic content and CO_3_ (TGA). Apatite Hb inversely correlated with Pb, while the degree of crystallinity positively correlated with Cu and Rb and negatively with Co and Sn (*p* < 0.005) (Appendix A).

With respect to TGA, its inverse correlations with XRD parameters have already been mentioned above. H_2_O, simple organic and total volatile content TGA parameters positively correlated with all FTIR-ATR parameters. Most TGA parameters positively correlated with CaO, Na_2_O, P_2_O_5_, SO_3_, SrO (chemical analysis), Ba, Co, Rb, Sn and Sr (ICP), while they inversely correlated with K_2_O, SiO_2_ (chemical analysis), Cu, La, Ni and Sb (ICP) (*p* < 0.005) (Appendix A).

Most FTIR-ATR and TGA parameters positively correlated with each other. In addition, FTIR-ATR parameters positively correlated with CaO, MgO, Na_2_O, P_2_O_5_, SO_3_, SrO (chemical analysis), Ba, Co, Sn and Sr (ICP), and negatively correlated with Fe_2_O_3_, K_2_O, SiO_2_ (chemical analysis), Cu, La and Ni (ICP) (*p* < 0.005) (Appendix A).

## 3. Discussion

The “bones and stones” concept is about finding similarities between osteology and geology/geophysics. We wished to determine whether analytic methods used in geology/geophysics to assess the composition and structure of minerals could be applied to bone research. We also wanted to compare inorganic and organic components in the bones with BMD as determined by QCT. In this preliminary pilot study, we applied chemical analysis, ICP-OES, ICP-MS, XRD, TGA and FTIR-ATR to analyse 14 bovine bone samples. The results of these physico-chemical methods were compared by QCT bone densitometry of total, trabecular and cortical bone [40]. Some bone studies have used XRD, FTIR or TGA [41,42,43,44], as well as other spectroscopy techniques [8,9,10,11,12,13,14,15]. However, we did not find studies where such a plethora of techniques was applied at the same time and where these techniques were combined with ICP-MS, ICP-OES and QCT.

Bulk major and trace element composition of adult bovine bone samples was determined by ICP-OES and ICP-MS. Our measurements indicated that the bones were, in principle, homogenous. Some contamination from the utensils used during the sample preparations can not be ruled out when evaluating Fe, Al, Ni, Sn and Pb concentrations. These elements were found in higher concentrations in the sample that was analysed first in the series, suggesting that the origin of this contamination may be the metal tools used during the sample preparation. Since the total for the chemical measurements accounted only for a ~65 wt%, it might well be assumed that apart from the inorganic/mineral components, the “missing” weight consists of the organic part of the bones (e.g., C, H, S, N, O). These organic components of the bones cannot be determined by the applied sample preparation and ICP techniques [4,28,30,38,39].

Among the mineral components of the bones, the XRD could only detect apatite. The exact type of the apatite could not be identified. The relatively narrow reflections characteristic of apatite implied that the apatite was highly crystallized, which suggested that the bones belonged to aged animals. Indeed, the mean age of bovine bone samples was 5 years. The bands with decreasing half-widths might indicate higher crystallinity of apatite crystals, which increases with the ageing of bones [49]. The half-width of the band at ~26° is particularly sensitive to the crystalline ordering of apatite [35].

The relatively low total inorganic/mineral content of the bones and the fact that the XRD could only detect apatite as a mineral component justified the conduction of additional TGA and FTIR-ATR assessments [32,33,34,35,36]. TGA determines the weight loss in parallel with increasing temperature. An attempt was made to determine the composition of carbonated hydroxyapatite from the amount of CO_3_ lost. However, this was complicated by the presence of poorly crystallized carbonates (i.e., calcite and dolomite) that could have also contributed to the released CO_2_ content [33]. According to the exothermic band (732 °C–799 °C), there might be a phase transition from apatite to β-tricalcium phosphate. Furthermore, it needs to be mentioned that the exothermic reaction does not happen in the bone samples treated with additional H_2_O_2_.

The total volatile content of the samples was about 35 wt%. This, together with the 65 wt% inorganic components determined by the chemical analysis, yields a total approaching 100 wt%. Since a large portion of the volatile content (H_2_O, CO_2_, etc.) is liberated from the organic components, it can be concluded that the inorganic/mineral to organic ratio of the examined bone samples was approximately 35:65. Furthermore, a small amount of carbonate might originate either from the apatite or the carbonate minerals, such as calcite or dolomite. This means that besides apatite, carbonate might also be present as an inorganic mineral constituent. Nonetheless, carbonates cannot be detected by XRD due to their poor crystallinity. The presence of calcite and/or dolomite is also supported by the considerable amount of Mg (~0.6 wt%) observed in the chemical composition analyses. This is because Mg is not a major constituent of either the apatite or the organic portion of the bone [1,22].

The chemical composition also suggests the presence of salts, as the Na content of the bone samples is relatively high (0.7 wt%). Na, as well as Mg, is not a major component of apatite, carbonate or the organic portion. The TGA analysis of the bone samples revealed that the missing 35 wt% mass can be considered organic matter in addition to apatite and carbonate. However, the TGA analysis did not get us closer to the composition of the organic matter in the bone samples. Therefore, samples were further investigated by FTIR-ATR.

The FTIR-ATR analysis revealed that the organic portion consisted of collagens containing amide groups, since their typical bands (OH, CH, CO, NC) could be clearly identified in the infrared spectra [34,35,36,50]. Furthermore, the area under the amide bands correlated well with the organic content determined by TGA. It was also observed that with the increasing content of the organic portion, the concentration of S also increased.

Our preliminary study has strengths and limitations. To the best of our knowledge, this might be one of the first studies applying numerous geological techniques (TGA, XRD, FTIR, ICP-MS/ICP-OES) in addition to QCT to bone research. We also logically applied the various methods step by step to determine the inorganic and organic content of the bones. A major strength of the study is that we compared these results with BMD as determined by QCT. Possible limitations include the limited number of bone samples, the use of bone powder and the possibility of contamination from the utensils and metal tools used during sample preparation. We used standard QCT and not high-resolution QCT. Moreover, our preliminary study is rather descriptive, yielding a few conclusions but without a deeper and complete picture of bone mineral composition. Further studies are needed to validate our results in human bones.

## 4. Methods and Materials

### 4.1. Specimen Preparation

A total of 14 adult male bovine tibial bone samples (B1–B14) were used for this study. The mean age of the animals was 1817 days (range: 515–3791 days). Bovine bones were obtained from the slaughterhouse. The midshaft region of the bones was used for the study. This midshaft region mostly contains cortical bone. Every bone sample was analysed individually. Before the analytic studies, the organic matter, including both meat and marrow, was removed from the bones mechanically in boiling water and treated with 1% hydrogen peroxide. The bones were cut into smaller pieces using a saw, and the samples were ground further to achieve a bone powder with particle sizes between 2 and 63 μm and to get roughly around 10–20 g of each sample. The phase analytical procedures, such as XRD (1 g), TG (100 mg) and FTIR-ATR (~5 mg), used the powdered bone, in the respective quantities, to determine their mineral and, to a smaller extent, organic components. For the chemical analysis, the powder (~3 g) was chemically dissolved with aqua regia facilitated by microwaves, and a clear homogenous solution was obtained. This solution was used to determine the major and trace element composition using the ICP-OES and ICP-MS techniques, respectively [27,34].

All analytical measurements above were implemented at the SARA Department of Geology, Budapest, Hungary. The study was approved by the Hungarian Scientific Research Council Ethical Committee (approval No. 37597-1/2012/EKU) on 21 January 2013.

### 4.2. Quantitative Computed Tomography

Peripheral QCT assessments of the bovine bone samples were performed using a Stratec XCT-2000 instrument (Stratec Medizintechnik GmbH, Pforzheim, Germany) as described before for human bones [40]. This was a QCT equipment with a higher voxel size and lower resolution and not a standard, high-resolution QCT. We assessed the epiphysis and diaphysis for trabecular (cancellous) and cortical bone BMD, respectively. Total, trabecular, and cortical volumetric (3-dimensional) BMD values, as determined by QCT, are expressed as mg/cm^3^. The applied setting to acquire the image was 0.59 mm voxel. The analysis was completed using the XCT6.00B software (Stratec Medizintechnik GmbH, Pforzheim, Germany).

### 4.3. XRD

Powdered bone samples underwent X-ray powder diffraction analysis using a Philips PW 1730 diffractometer (Philips, Amsterdam, Netherlands) (copper cathode, 40 kV and 30 mA tube–current, graphite monochromator, goniometer speed 2°/min). A semi-quantitative assessment of the relative concentrations of the phases was performed by relative intensity ratios and full width at half maximum (FWHM) of specific reflections of minerals using XDB Powder Diffraction Phase Analytical software 2.7 [29,30]. The assessed parameters were apatite peak intensity (arbitrary units, AU), the ratio of peak intensities of crystallographic planes at 211 and 222, that at 211 and 300, the degree of crystallinity (%) and the c-axis (Å) [29,30,51].

Below are the details of the XRD technique:
Apatite peak intensity is the height of the strongest peak of apatite from the baseline.The ratio of peak intensity at 211 and 222 is the measurement ratio of peak 211 and 222 intensities from the baseline.The ratio of peak intensity at 211 and 300 is the measurement ratio of peak 211 and 300 intensities from the baseline.The degree of crystallinity was calculated using diffraction peaks 112 and 300) of apatite with the equation of Landi et al. [51] given as:Xc = 1 − (V_112/300_/I_300_),

where I_300_ is the 300 intensity reflection and V_112/300_ is the intensity of the hollow between 112 and 300 reflections.
The c-axis parameter represents the length of the c-axis of apatite. Using standard Bragg’s Law, the interplanar spacing d_002_ was calculated from the diffraction angle of the 002 reflection. The c-axis length (in Å) was then determined by multiplying d_002_ by 2.

### 4.4. TGA

Thermogravimetric analysis (TGA) was performed using a Derivatograph–PC. Temperature was gradually adjusted from room temperature (20 °C) to 1000 °C (10 °C/min). Al_2_O_3_ was used as inert material. A total of 100 mg of the samples was heated in a ceramic crucible. The quantitative determination of the thermally active minerals was based on stoichiometric calculations of the decomposition processes of the identified minerals due to loss of mass during heating [31,32,33]. Molecular water, simple and composite organic minerals, carbonate (CO_3_) content, and total and >900 °C volatile matters were measured (wt%) [33]. From the mass change in each reaction and with the knowledge of the thermochemical reaction equation, the mass percent ratio can be determined for the mineral component in the sample. The stoichiometric factor introduced for the quantitative determination is as follows:

f = M/m, where M is the molecular mass of the mineral and m is the mass change during the given reaction [33].

### 4.5. FTIR-ATR

Powdered bone samples were investigated by a Fourier-transform infrared spectrometer (Bruker Co., Billerica, MA, USA) (Bruker Vertex 70) coupled to a single pass ATR cell (Platinum ATR) in the mid-infrared spectral region (400–4000 cm^−1^) as described before [34]. For the analysis, a liquid nitrogen cooled MCT detector was used. Each spectrum displays spectral features of the diamond ATR crystal between ~1800 and 2200 cm^−1^. All samples were placed in roughly equal amounts (3–5 mg) in polished glass containers and heated for at least 30 min at 80 °C directly before the measurements. Glass containers were closed with proper sealing ground glass lids and wrapped in parafilm immediately after heating. This procedure is able to remove the majority of the absorbed atmospheric water that could modify absorption characteristics, and can prevent aggregation of mineral particles.

For the samples and their respective backgrounds, 64 scans were acquired at a nominal resolution of 4 cm^−1^. The sample powder is pressed with constant pressure on a small diamond crystal plate, which allows obtaining reproducible spectra from different sample batches. Baseline and extended ATR corrections were performed by applying the OPUS 6.5 software package for processing the spectra before evaluation. For the advanced ATR correction 1 reflection, 45° of incidence angle and a 1.54 refractive index were assumed [34].

The concentration of certain components of bioapatite was semi-quantitatively estimated based on the integrated area under their characteristic absorption bands (cm^−1^). The molecular water, hydroxyl and alkyl (H_2_O–OH^−^ and CH_3_) components were estimated based on the bands between 2688 and 3710 cm^−1^. For the alkyl (CH_3_, organic) component, only the band between 2825 and 3004 cm^−1^ is considered. The carbonate + phosphate (${\mathrm{CO}}_{3}^{2-}+{\mathrm{PO}}_{4}^{3-}$), carbonate ${\mathrm{CO}}_{3}^{2-}$, amid + carbonate ${\mathrm{CO}}_{3}^{2-}$ and amid I contents of bio-apatite were estimated using the integrated area between the following wavenumbers: 853–1166, 853–890, 1190–1800 and 1593–1717 cm^−1^, respectively [35,36,50]. The extent of mineralisation was estimated based on the amid I/(phosphate + carbonate (${\mathrm{CO}}_{3}^{2-}+{\mathrm{PO}}_{4}^{3-}$) ratio. This ratio is high when there is more organic material present and lower when the proportion of the crystalline bioapatite component is higher. The carbonate/(carbonate + phosphate) [${\mathrm{CO}}_{3}^{2-}$/$({\mathrm{CO}}_{3}^{2-}+{\mathrm{PO}}_{4}^{3-}$)] ratio measures the degree of carbonate substitution into the bioapatite structure. The carbonate group can substitute both the hydroxyl (OH^−^, referred to as type A) and phosphate (${\mathrm{PO}}_{4}^{3-}$ referred to as type B) components, both of which have important structural and biological consequences [52,53].

### 4.6. Chemical Analysis

To determine the main element (oxide) content of the bones, 0.25 g of the samples were digested in a Milestone 1200 Mega microwave unit with aqua regia (a mixture of nitric acid and hydrochloric acid, optimally in a molar ratio of 1:3) by microwave heating and the solutions were filled to 50 mL.

Si, Al, Ca, Mg, Na, K, Fe, Mn, P, S, Ba, Sr and Ti are considered ‘major elements’ in geology/geochemistry. These elements were analysed from the bone sample solutions (except Al) by ICP-OES. The analysis was performed on a Jobin Yvon ULTIMA 2C ICP-OES spectrometer, equipped with both a monochromator and a polychromator. Certified synthetic standard solutions were used for the calibration, and certified reference rock materials (GBW07109–GBW07114, Ministry of Geology and Mineral Resources, Beijing, China) were applied as quality check (QC) samples. The results were given in element concentration (g/kg, mg/kg) as well as in oxide form (in wt%).

The minor and trace elements’ content was analysed with ICP-OES and ICP-MS. We used ICP-OES to determine As, Cu, Li and Zn content and ICP-MS to evaluate Ba, Bi, Cd, Ce, Co, Cr, Cs, Cu, La, Mn, Mo, Nd, Ni, Pb, Pr, Rb, Sb, Sn, Sr, Th, Tl, U, Y and Zn content in the bone samples (mg/kg). The ICP-OES analysis was performed on a Jobin Yvon ULTIMA 2C ICP-OES spectrometer. The ICP-MS assessment was carried out using Perkin Elmer ELAN DRC II equipment. For calibration, certified synthetic standard solutions were applied [34].

### 4.7. Statistical Analysis

The statistical analysis was conducted by SPSS 26.0 (IBM, Armonk, NY, USA) software. Data are expressed as the mean ± SD. Simple correlations were determined by Pearson’s analysis. In addition, univariable and multivariable regression analyses using the stepwise method were used to determine independent correlations between any two variables. The β coefficients showing linear correlations between two parameters were calculated. The B (+95% CI) regression coefficient showed independent correlations between dependent and independent variables over time. We considered *p* values < 0.05 significant in all statistical tests.

## 5. Conclusions

Based upon a collaboration between medical and geophysical personnel, here we performed a geophysical/geochemical analysis of bones using various techniques. With the applied pre-analytical treatments and baseline analytical procedures, we were able to define the total composition of the studied bone samples. When asking a medical question in bone research, first, it should be determined whether the whole bone or only the inorganic/mineral content is of interest. If only the latter, further methodological development may be envisaged. The inorganic/mineral content of the examined bones was carbonate-apatite, carbonates and salts (~65 wt%), whereas the organic part was composed of collagens (~35 wt%). For the exact definition of the apatite and the salts, it would be necessary to carry out an electron microprobe analysis (EMPA). From a medical point of view, the composition of the studied bones could be reliably examined, with the exception of As and, due to suspicion of contamination, Fe, Al, Ni, Sn and Pb. This was a preliminary pilot study carried out on bovine bone samples. This study might be followed by similar investigations on human bone samples.

## Figures and Tables

**Figure 1 ijms-25-10733-f001:**
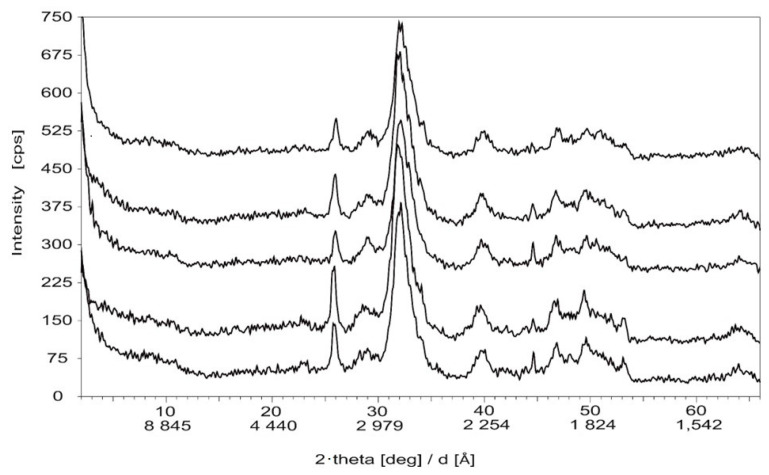
X-ray diffraction (XRD). A representative graph of five randomly selected bovine bone samples.

**Figure 2 ijms-25-10733-f002:**
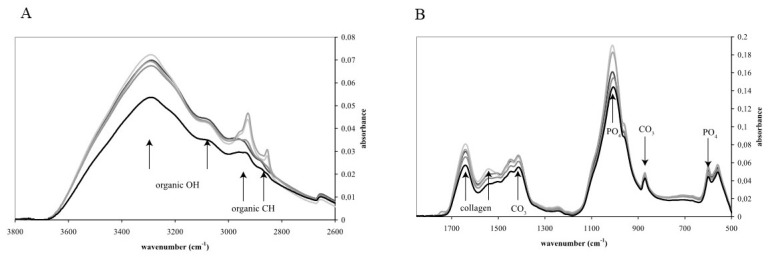
Fourier-transform infrared spectrometry with attenuated total reflectance accessory (FTIR-ATR). A representative graph of five randomly selected bovine bone samples. (**A**,**B**) are for the same sample. (**A**) shows peaks for 2600 to 3800, while (**B**) is for 500–1700 wavenumbers.

**Table 1 ijms-25-10733-t001:** Descriptive statistics.

Parameter	Mean ± SD (Range)
Age	1816.6 ± 1093.6 (515–3791)
QCT	
Total BMD (mg/cm^3^)	954.5 ± 59.3 (832.9–1031.3)
Trabecular BMD (mg/cm^3^)	647.3 ± 114.9 (418.1–795.5)
Cortical BMD (mg/cm^3^)	1205.0 ± 22.1 (1172.1–1247.2)
Total area (mm^2^)	1154.9 ± 146.8 (909.6–1432.1)
Trabecular area (mm^2^)	519.4 ± 66.0 (409.0–644.3)
Cortical area (mm^2^)	635.2 ± 80.6 (500.6–787.8)
Chemistry	
CaO (wt%)	38.15 ± 1.52 (36.09–41.35)
Fe_2_O_3_ (wt%)	0.01 ± 0.01 (0.003–0.04)
K_2_O (wt%)	0.04 ± 0.01 (0.02–0.06)
MgO (wt%)	0.62 ± 0.03 (0.56–0.69)
Na_2_O (wt%)	0.83 ± 0.05 (0.75–0.93)
P_2_O_5_ (wt%)	28.02 ± 1.30 (25.57–29.78)
SO_3_ (wt%)	0.18 ± 0.01 (0.17–0.20)
SiO_2_ (wt%)	0.06 ± 0.06 (0.01–0.20)
SrO (wt%)	0.02 ± 0.002 (0.02–0.03)
ICP-OES	
Cu (mg/kg)	8.76 ± 7.87 (1.91–33.01)
Li (mg/kg)	1.99 ± 0.53 (1.21–2.84)
Zn (mg/kg)	60.87 ± 18.32 (36.76–110.45)
ICP-MS	
Ba (mg/kg)	50.83 ± 13.13 (26.35–70.42)
Ce (mg/kg)	0.16 ± 0.10 (0.05–0.38)
Co (mg/kg)	0.61 ± 0.09 (0.38–0.74)
Cr (mg/kg)	1.10±0.37 (0.68–1.73)
Cu (mg/kg)	8.27 ± 6.30 (1.95–26.62)
La (mg/kg)	0.08 ± 0.04 (0.05–0.18)
Mn (mg/kg)	6.21 ± 9.75 (0.82–39.89)
Mo (mg/kg)	0.36 ± 0.31 (0.15–1.44)
Ni (mg/kg)	1.41 ± 0.59 (0.86–3.07)
Pb (mg/kg)	0.83 ± 0.63 (0.31–2.81)
Rb (mg/kg)	0.23 ± 0.15 (0.09–0.57)
Sb (mg/kg)	0.07 ± 0.03 (0.05–0.13)
Sn (mg/kg)	8.18 ± 4.42 (0.47–12.37)
Sr (mg/kg)	209.83 ± 32.72 (146.16–261.08)
Zn (mg/kg)	61.17 ± 18.44 (37.09–111.41)
XRD	
Apatite peak intensity (AU)	1590 ± 70 (1460–1690)
Ratio of 211 and 222 peak intensity	5.10 ± 0.31 (4.82–5.79)
Ratio of 211 and 300 peak intensity	1.77 ± 0.09 (1.61–1.95)
Degree of cristallinity (%)	0.01 ± 0.01 (0–0.05)
C-axis (Å)	6.89 ± 0.01 (6.87–6.90)
Thermogravimetric analysis (TGA)	
H_2_O (wt%)	6.94 ± 0.66 (6.20–8.07)
Simple organic content (wt%)	11.03 ± 1.12 (8.84–12.50)
Composite organic content + OH^−^ (wt%)	9.08 ± 1.24 (6.88–10.87)
CO_3_ content (wt%)	1.16 ± 0.15 (0.98–1.52)
Total volatile content (wt%)	28.56 ± 3.01 (23.05–32.93)
FTIR-ATR	
H_2_O + CH (cm^−1^)	10.07 ± 1.46 (7.99–13.98)
CH (cm^−1^)	0.20 ± 0.02 (0.16–0.24)
PO_4_ + CO_3_ (cm^−1^)	8.90 ± 0.95 (7.24–10.21)
CO_3_ (cm^−1^)	0.10 ± 0.02 (0.06–0.12)
Amide I + CO_3_ (cm^−1^)	7.65 ± 0.78 (6.38–8.94)
Amide I (cm^−1^)	1.36 ± 0.18 (1.09–1.81)
Amide I/(PO_4_ + CO_3_) ratio	6.61 ± 0.59 (5.45–7.41)
CO_3_/(PO_4_ + CO_3_) ratio	0.01 ± 0.00 (0.01–0.01)

**Table 2 ijms-25-10733-t002:** Correlations between QCT BMD and physico-chemical parameters.

	QCT Parameters		
	Total BMD (mg/cm^3^)	Trabecular BMD (mg/cm^3^)	Cortical BMD (mg/cm^3^)
Thermogravimetric Analysis			
H_2_O	R = 0.533		R = 0.585
	*p* = 0.049		*p* = 0.028
Simple organic content	R = 0.631	R = 0.612	
	*p* = 0.015	*p* = 0.020	
Composite organic content	R = 0.664	R = 0.647	
	*p* = 0.010	*p* = 0.012	
Total volatile content	R = 0.661	R = 0.631	R = 0.546
	*p* = 0.010	*p* = 0.015	*p* = 0.043
FTIR-ATR			
H_2_O + CH	R = 0.613	R = 0.660	
	*p* = 0.020	*p* = 0.010	
CH	R = 0.620	R = 0.612	
	*p* = 0.018	*p* = 0.020	
PO_4_ + CO_3_	R = 0.540		R = 0.656
	*p* = 0.046		*p* = 0.011
CO_3_	R = 0.645	R = 0.603	R = 0.576
	*p* = 0.013	*p* = 0.022	*p* = 0.031
Amide I + CO_3_	R = 0.705	R = 0.704	
	*p* = 0.005	*p* = 0.005	
Amide I	R = 0.694	R = 0.682	
	*p* = 0.006	*p* = 0.007	
CO_3_/(PO_4_ + CO_3_)	R = 0.671	R = 0.665	
	*p* = 0.009	*p* = 0.009	
Chemistry			
CaO	R = 0.664	R = 0.606	R = 0.688
	*p* = 0.001	*p* = 0.022	*p* = 0.007
Fe_2_O_3_			R = −0.544
			*p* = 0.045
MgO			R = 0.625
			*p* = 0.017
Na_2_O			R = 0.571
			*p* = 0.033
SiO_2_	R = −0.621	R = −0.569	R = −0.596
	*p* = 0.018	*p* = 0.034	*p* = 0.024
ICP-MS			
Co	R = 0.568		R = 0.656
	*p* = 0.034		*p* = 0.011
Mn			R = 0.535
			*p* = 0.049
Mo			R = 0.621
			*p* = 0.018
Ni	R = −0.651	R = −0.615	R = −0.534
	*p* = 0.012	*p* = 0.019	*p* = 0.049
Sn	R = 0.608	R = 0.613	
	*p* = 0.021	*p* = 0.020	

**Table 3 ijms-25-10733-t003:** Univariable and multivariable analysis of determinants of quantitative CT results.

Dependent Variable	Independent Variable	Univariable Analysis				Multivariable Analysis			
		β	*p*	B	95% CI	β	*p*	B	95% CI
Total BMD	Age	0.717	0.004	0.040	(0.016–0.065)	0.644	<0.001	0.036	(0.021–0.052)
	Thermogravimetric Analysis								
	H_2_O	0.533	0.049	47.680	(0.019–95.197)				
	Simple organic content	0.631	0.015	33.582	(7.635–59.529)				
	Composite organic content	0.664	0.010	31.704	(9.273–54.135)				
	Total volatile content	0.661	0.010	13.047	(3.741–22.354)				
	FTIR-ATR								
	H_2_O + CH	0.613	0.020	24.864	(4.704–45.025)				
	CH	0.620	0.018	1638.8	(334.3–2943.4)				
	PO_4_ + CO3	0.540	0.046	33.841	(0.680–67.003)				
	CO_3_	0.645	0.013	1965.8	(499.3–3432.2)				
	Amide I + CO_3_	0.705	0.005	53.530	(19.691–87.369)				
	Amide I	0.694	0.006	229.13	(79.76–378.50)				
	CO_3_/(PO_4_ + CO_3_)	0.671	0.009	31515.5	(9637.6–53393.4)				
	ICP-MS								
	Co	0.568	0.034	363.83	(31.91–695.74)				
	Ni	−0.651	0.012	−65.893	(−114.230–−17.557)	−0.569	0.001	−57.558	(−85.326–−29.790)
	Sn	0.608	0.021	8.171	(1.468–14.873)				
	Chemistry								
	CaO	0.664	0.010	25.991	(7.606–44.376)				
	SiO_2_	−0.621	0.018	−625.88	(−1122.2–−129.53)				
Trabecular BMD	Age	0.706	0.005	0.077	(0.028–0.126)	0.638	0.001	0.070	(0.035–0.104)
	Thermogravimetric Analysis								
	Simple organic matter	0.612	0.020	63.055	(11.782–114.33)				
	Composite organic matter	0.647	0.012	59.851	(15.533–104.17)				
	Total volatile material	0.631	0.015	24.128	(5.484–42.772)				
	FTIR-ATR								
	H_2_O + CH	0.660	0.010	51.877	(14.740–89.013)				
	CH	0.612	0.020	3135.5	(588.47–5682.5)				
	CO_3_	0.603	0.022	3561.9	(596.85–6526.9)				
	Amide I + CO_3_	0.704	0.005	103.45	(37.716–169.17)				
	Amide I	0.682	0.007	436.07	(141.98–730.16)				
	CO_3_/(PO_4_ + CO_3_)	0.665	0.009	60518.7	(17825.6–103211.9)				
	ICP-MS								
	Ni	−0.615	0.019	−120.57	(−217.87–−23.267)	−3.722	0.003	−104.58	(−166.43–−42.735)
	Sn	0.613	0.020	15.948	(3.020–28.876)				
	**Chemistry**								
	CaO	0.606	0.022	45.891	(7.959–83.822)				
	SiO_2_	−0.569	0.034	−1109.7	(−2119.3–−100.12)				
**Cortical BMD**	Thermogravimetric analysis								
	H_2_O	0.585	0.028	19.494	(2.482–36.506)				
	Simple organic matter								
	Composite organic matter								
	Total volatile material	0.546	0.043	4.018	(0.142–7.895)				
	FTIR-ATR								
	PO_4_ + CO3	0.656	0.011	15.336	(4.248–26.424)				
	CO_3_	0.576	0.031	655.41	(70.747–1240.1)				
	ICP-MS								
	Co	0.656	0.011	156.75	(43.220–270.28)				
	Mo	0.621	0.018	44.512	(9.217–79.807)				
	Ni	−0.534	0.049	−20.147	(−40.230–−0.063)				
	Chemistry								
	CaO	0.688	0.007	10.037	(3.378–16.695)	0.564	0.008	8.226	(2.616–13.837)
	MgO	0.625	0.017	417.81	(89.267–746.35)	0.478	0.019	319.94	(62.685–577.19)
	Na_2_O	0.571	0.033	253.05	(24.197–481.91)				
	SiO_2_	−0.596	0.024	−224.07	(−413.71–−34.433)				

## Data Availability

Original data can be obtained from the main investigator upon request.

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
