# Peer review of "Bones or Stones: How Can We Apply Geophysical Techniques in Bone Research?"

_ijms, 2024, doi:10.3390/ijms251910733_

Round 1

Reviewer 1 Report

Comments and Suggestions for Authors

The study by Szekanecz explores the chemical and mineralogical composition of bovine tibial bone samples using different techniques. These techniques include X-ray diffraction, thermogravimetry, Fourier transformed infrared spectrometry, inductively coupled mass spectrometry (ICP-MS), and optical emission spectrometry (ICP-OES). Additionally, quantitative CT (QCT) was used to assess bone mineral density (BMD). The results showed that the bone composition, including major and trace elements, could be effectively determined. The study concluded that these geological techniques, combined with QCT, provide a reliable method for analyzing bone composition, and could be useful in medical research.

Although the paper is overall well-written, I have some issue regarding the relevance of the paper. The authors state throughout the paper that the methods employed are “geological techniques”, but it has been decades that they are used for bone analysis. If the main point was to show that ICP might be a more useful methods, then it should be rewritten a bit more into this direction. However, in this case, studies have already been published. The comparison with BMD in this study might not be very useful given the parameter used for the CT scans.

Below are few more specific comments:

Introduction:

-       First paragraph: Bone structure has been assessed with much more techniques, from the nanoscale to the microscale (SEM etc…). Are the authors referring to in-vivo only? If yes, this should be clearly mentioned.

-       The first paragraph of the introduction referred to the fact that bone biopsy is needed to assess bone structure but limit the ability to fully study the effects of anti-osteoporotic drugs on bone health. How is it related?

-       A lot of methods are also available for spectroscopy in bone as it is mentioned later in the introduction, but comprises also RAMAN, EDX, APT and so on. For trace elements, these techniques, atomic absorption spectroscopy, mass spectroscopy, X-ray fluorescence, etc have also been used. To highlight the importance of ICP methods, it would be relevant to explain and compare a bit more the methods with advantages/disadvantages. Plethora of literature with bone samples exists within each of those techniques.

-       line 113: what would be the advantages to apply all the techniques? It is not clear if the aim is to compare techniques…In addition, Raman & BMD for instance have been employed, how does the results compare to these one?

Materials & Methods:

-       Overall, this part is missing details on samples preparation:

-       Was cortical vs. cancellous bone separated? If yes, how was it sampled?

-       Which part of the bone was used (midshaft, epiphysis, etc.)

-       What age was the bone? (mature vs. juvenile?)

-       2.2: Normally, to avoid quantification errors, BMD is evaluated on images with a pixel (voxel for volumetric) size ranging from 1 to 20 um app. Here more than 500 um are used which can greatly affect the results.

-       which software was used for quantification? How was it done?

-       why using single slice QCT and not volumetric?

Results:

-       3.1: Once again, the results obtained for the QCT might not be reliable; both because of the voxel size but also because the location of sampling is not clearly stated.

-       line 221 to 223: I am not sure to understand the relevance of the comparison with a human historical cohort where the age and pathological state is not mentioned. I would rather delete this statement and compare with literature on bovine BMD.

Discussion:

-       The interpretation of the XRD and TGA results, particularly regarding the transition from apatite to β-tricalcium phosphate and the exothermic reactions observed, is somewhat speculative. Based on the result obtained, there are no evidence to fully support these interpretations.

-       The direct comparison of BMD result and the different techniques used is not obvious. What does it mean for the bone structure, etc.?

-       Here, the study uses bovine bones, which are not the same bone type compared to human bones, both in terms of composition and structure. While the authors acknowledge this in part, it does not fully address the implications for translating these results to human bones. In addition, the full potential of the techniques is not clearly highlighted: would the use of ICP be better, faster, etc.? How to control for contaminant? (the latter being a huge drawback if the result cannot be fully trusted)

The study's application of geological techniques to bone research is commendable, and it provides foundation for future work. However, the limitations, particularly the contamination risks, incomplete analysis, and not reliable results mean that its findings should be interpreted cautiously. If contamination controls could be provided and higher resolution QCT as well as a deeper interpretation of the results, the study would be greatly improved.

Author Response

The study by Szekanecz explores the chemical and mineralogical composition of bovine tibial bone samples using different techniques. These techniques include X-ray diffraction, thermogravimetry, Fourier transformed infrared spectrometry, inductively coupled mass spectrometry (ICP-MS), and optical emission spectrometry (ICP-OES). Additionally, quantitative CT (QCT) was used to assess bone mineral density (BMD). The results showed that the bone composition, including major and trace elements, could be effectively determined. The study concluded that these geological techniques, combined with QCT, provide a reliable method for analyzing bone composition, and could be useful in medical research.

Although the paper is overall well-written, I have some issue regarding the relevance of the paper. The authors state throughout the paper that the methods employed are “geological techniques”, but it has been decades that they are used for bone analysis. If the main point was to show that ICP might be a more useful methods, then it should be rewritten a bit more into this direction. However, in this case, studies have already been published. The comparison with BMD in this study might not be very useful given the parameter used for the CT scans.

We thank the reviewer for finding our manuscript well-written and we understand the issues mentioned by the reviewer. We try to address all issues raised by the Reviewer and make changes in the text accordingly in RED colour.

Below are few more specific comments:

Introduction:

-       First paragraph: Bone structure has been assessed with much more techniques, from the nanoscale to the microscale (SEM etc…). Are the authors referring to in-vivo only? If yes, this should be clearly mentioned.

Yes, we refer to in vivo only as we wished to point out that many techniques might have limitations in vivo. This is now added to the first paragraph.

-       The first paragraph of the introduction referred to the fact that bone biopsy is needed to assess bone structure but limit the ability to fully study the effects of anti-osteoporotic drugs on bone health. How is it related?

Indeed, the effects of drugs are not related to the topic of this paper, so this sentence is now removed.

-       A lot of methods are also available for spectroscopy in bone as it is mentioned later in the introduction, but comprises also RAMAN, EDX, APT and so on. For trace elements, these techniques, atomic absorption spectroscopy, mass spectroscopy, X-ray fluorescence, etc have also been used. To highlight the importance of ICP methods, it would be relevant to explain and compare a bit more the methods with advantages/disadvantages. Plethora of literature with bone samples exists within each of those techniques.

In our study, we did not want to put ICP into focus, this is just one technique we used among others. However, we fully agree with the reviewer and added some literature on other types of spectroscopy as suggested.

-       line 113: what would be the advantages to apply all the techniques? It is not clear if the aim is to compare techniques…In addition, Raman & BMD for instance have been employed, how does the results compare to these one?

As now pointed out even more strongly in the Introduction, the novelty of our study is that while many other studies allpied 2-3 techniques maximum at one point, almost no studies used such a plethora of multiple techniques at one time. Also, we did not find any studies when geophysical techniques were applied in conjunction with BMD assessment by QCT. We now cite studies where Raman and other types of spectroscopy were applied, however, as we performed our studies with different techniques and under different conditions, comparison was not feasible.

Materials & Methods:

-       Overall, this part is missing details on samples preparation:

Actually there was a full paragraph on specimen preparation. “Before the analytic studies, the organic matter including both meat and marrow was removed from the bones mechanically in boiling water and treatment with 1% hydrogen peroxide. The bones were cut into smaller pieces using a saw, and the samples then were ground further to achieve a bone powder with particle sizes between 2 and 63 μm to get roughly around 10-20 grams of each sample. The phase analytical procedures, such as XRD (1 g), TG (100 mg) and FTIR-ATR (~5 mg) used the powdered bone, in the respective quantities, to determine their mineral, and to a smaller extent, organic components. For the chemical analysis, the powder (~3 grams) was chemically dissolved with „Aqua Regia” facilitated by microwaves and a clear homogenous solution was obtained. This solution was used to determine the major and trace element compositions using the ICP-OES and ICP-MS techniques, respectively.”

-       Was cortical vs. cancellous bone separated? If yes, how was it sampled?

The midshaft region of the tibial bone was assessed, which predominantly consists of cortical bone. The tibias of the sacrificed animals were dissected from the cadaver, and the midshaft of the long bone was cut into smaller pieces using a saw. Thus, cancellous and cortical bones were not separated, but the midshaft had mostly cortical bone. On the other hand, the peripheral QCT assessed both trabecular and cortical bone. This information is now added to the methods section.

-       Which part of the bone was used (midshaft, epiphysis, etc.)

The midshaft region of the tibia bone was used. This information is now added.

-       What age was the bone? (mature vs. juvenile?)

The sacrificed animals were mature, on average 1817 days old (range: 515-3791 days). This information is now added.

-       2.2: Normally, to avoid quantification errors, BMD is evaluated on images with a pixel (voxel for volumetric) size ranging from 1 to 20 um app. Here more than 500 um are used which can greatly affect the results.

We agree, however the software was provided with the Stratec QCT instrument.

-       which software was used for quantification? How was it done?

As already written in this section, “analysis was completed using the XCT6.00B software (Stratec Medizintechnik GmbH, Pforzheim, Germany).” The software was provided with the instrument.

-       why using single slice QCT and not volumetric?

We are sorry that was an error, our QCT was able for volumetric BMD determination. We now corrected this.

Results:

-       3.1: Once again, the results obtained for the QCT might not be reliable; both because of the voxel size but also because the location of sampling is not clearly stated.

As presented above we assessed the midshaft region. Also the very same QCT was used for our human studies before (see ref 40 in the paper). The voxel size was provided by the manufacturer.

-       line 221 to 223: I am not sure to understand the relevance of the comparison with a human historical cohort where the age and pathological state is not mentioned. I would rather delete this statement and compare with literature on bovine BMD.

We deleted this comparison as requested. However, due to the lack of studies using similar conditions, we were unable to compare our results with literature on bovine BMD.

Discussion:

-       The interpretation of the XRD and TGA results, particularly regarding the transition from apatite to β-tricalcium phosphate and the exothermic reactions observed, is somewhat speculative. Based on the result obtained, there are no evidence to fully support these interpretations.

Yes, we agree that this statement has not been fully supported. Therefore, we reworded this statement as being speculative. Exothermic reactions were observed in TGA. XRD is of course unable to detect apatite to β-tricalcium phosphate transition. XRD only detected apatite in the samples. Also, the XRD statement is toned down by using “might”.

-       The direct comparison of BMD result and the different techniques used is not obvious. What does it mean for the bone structure, etc.?

We did not want to do more but correlate the different geophysical techniques assessing bone structure and content with BMD and we just used QCT instead of DXA for BMD assessment as this was available in the department. It might not be obvious but we simply tried to link composition with BMD.

-       Here, the study uses bovine bones, which are not the same bone type compared to human bones, both in terms of composition and structure. While the authors acknowledge this in part, it does not fully address the implications for translating these results to human bones. In addition, the full potential of the techniques is not clearly highlighted: would the use of ICP be better, faster, etc.? How to control for contaminant? (the latter being a huge drawback if the result cannot be fully trusted)

We have already performed similar studies on human bones, but this will be in a future paper, so we did not want to speculate on this. We already indicated that this was a pilot study and human studies might follow in the last sentence of the Conclusion. So now we add a sentence on this issue. We demonstrated a logical sequence of the techniques in the text. ICP is not better or faster, it yields to different dimensions than TGA, XRD, etc. ICPs were used for the first step of screening, followed by TGA, XRD and FTIR. The possibility of contamination is discussed as a limitation of the study. It is hard to control for this as a standardized procedure was used.

The study's application of geological techniques to bone research is commendable, and it provides foundation for future work. However, the limitations, particularly the contamination risks, incomplete analysis, and not reliable results mean that its findings should be interpreted cautiously. If contamination controls could be provided and higher resolution QCT as well as a deeper interpretation of the results, the study would be greatly improved.

Many thanks for this summary. Unfortunately, this study cannot be repeated using different conditions mentioned by the Reviewer, however, these conditions are now added as limitations in the Discussion. Moreover, we use these suggestions when performing a human study.

Reviewer 2 Report

Comments and Suggestions for Authors

General comment: the manuscript has some value, but in its current form, I do not recommend it for publication. Significant revisions are required, and the key points for improvement are outlined below.

Introduction: The use of the term 'geophysical techniques' seems inappropriate for the methods described (FTIR, TGA, ICP-MS, etc.), as these are primarily chemical and physical techniques used across multiple scientific disciplines, including chemistry, physics, and biology. What is more, the manuscript lacks a clearly defined research hypothesis and objective. The purpose of the study is not explicitly stated, making it difficult to understand the study's contribution to the field. It also  does not clearly define the intended audience. Given the interdisciplinary nature of the techniques discussed (geological methods, bone research, and medical applications), it is important to specify whether the target readers are medical researchers, geologists, or multidisciplinary scientists. The techniques mentioned, such as FTIR, TGA, ICP-MS, and ICP-OES, have been widely used in preclinical studies for years, particularly in the context of bone research. Therefore, the novelty and unique contribution of this study should be more clearly articulated.

Specimen preparation: Please provide details regarding the origin of the bones and the age of the animals used in the study, as this is essential for interpreting the results. How were the bones stored? Were they dried or equilibrated for humidity before the analyses? Did the procedure involve the entire bones from the bovine samples? Given the size of bovine bones, this seems unlikely. Please specify which part of the bone was used for analysis and whether the samples were pooled or analyzed individually. This clarification is crucial, as later in the manuscript you discuss the analysis of trabecular and cortical bone, which differ significantly in terms of bone turnover, mineral content (including all macro- and micro-minerals, not just calcium and phosphorus), and bulk density. Thus, this clarification is especially important chemical analysis studies ICP-OES and ICP-MS.

Methods: Providing the explicit formulas used in the calculations would significantly enhance the value of this work. If the authors' intention is to create a reference work showcasing the advantages of 'geophysical' methods in bone analysis, I strongly recommend including a detailed description of how the specific parameters measured by XRD and TA techniques were calculated, similar to what was done for FTIR. Rather than simply stating 'according to standard procedures,' offering clear access to all necessary formulas and mathematical models would ensure that other researchers can accurately replicate these bone analyses in their future work.

Qct: Please provide precise information regarding the specific areas of the bone that were analyzed—where exactly was the cortical bone assessed (e.g., epiphysis, diaphysis, or metaphysis, and in which specific regions), and which region was chosen for the trabecular bone (e.g., distal or proximal epiphysis, metaphysis, or diaphysis)? Additionally, it should be clearly stated in the introduction that this study correlates the results of 'geophysical' methods with Peripheral Quantitative Computed Tomography (pQCT), not conventional QCT. This distinction is important, as pQCT, with a significantly lower resolution (0.59 mm voxel size in the present study), differs from the high resolution of standard QCT (potentially up to 0.01 mm), which limits the ability to accurately determine parameters such as trabecular thickness. In turn, the advantage of pQCT is its applicability in in vivo studies on the human forearm, as it is a rapid and non-invasive technique.

The Methods section lacks a 'Statistical Analysis' subsection, which is essential since correlation, and univariable and multivariable regression analyses appear to be a key components of this work. Please include detailed information on the statistical methods and tests used to evaluate the data.

L215: Does 'age' refer to the age of the animals or the duration for which the isolated bones were stored before analysis?

L298: When describing correlations (a strictly defined statistical term), avoid using the word 'associations.' These terms are not interchangeable, and 'correlation' should only be used when referring to statistically verified relationships.

The conclusions section needs revision, but this can be properly addressed once the precise research hypothesis and the aim of the study are clarified.

Author Response

General comment: the manuscript has some value, but in its current form, I do not recommend it for publication. Significant revisions are required, and the key points for improvement are outlined below.

We thank the Reviewer for the thorough revision and finding some value in the paper. We tried to improve our manuscript according to the recommendations and comments by the Reviewers.

Introduction: The use of the term 'geophysical techniques' seems inappropriate for the methods described (FTIR, TGA, ICP-MS, etc.), as these are primarily chemical and physical techniques used across multiple scientific disciplines, including chemistry, physics, and biology. What is more, the manuscript lacks a clearly defined research hypothesis and objective. The purpose of the study is not explicitly stated, making it difficult to understand the study's contribution to the field. It also does not clearly define the intended audience. Given the interdisciplinary nature of the techniques discussed (geological methods, bone research, and medical applications), it is important to specify whether the target readers are medical researchers, geologists, or multidisciplinary scientists. The techniques mentioned, such as FTIR, TGA, ICP-MS, and ICP-OES, have been widely used in preclinical studies for years, particularly in the context of bone research. Therefore, the novelty and unique contribution of this study should be more clearly articulated.

Now we changed the term geological to geophysical/geochemical throughout the paper. We now also explain, as most techniques have been used, we just call it a Bones and stones project as rheumatologists and geophysicists built up a nice collaboration for this study. The geophysical and geochemical techniques were performed in the Geophysical Institute. As this is a paper with relevance for bone research and osteoporosis, of course rheumatologists and other medical disciplines are the target, not the geologists. We now added a lot more info that these techniques, as well as other types of spectroscopy have indeed been used before. We just point out that in most studies only one or two of these techniques were applied, while we used a large plethora of these techniques. Also, the novelty of our study is that for the first time we connected many geophysical/geochemical techniques with BMD assessments by QCT. All this information is now added to different parts of the text in RED colour.

Specimen preparation: Please provide details regarding the origin of the bones and the age of the animals used in the study, as this is essential for interpreting the results. How were the bones stored? Were they dried or equilibrated for humidity before the analyses?

Bone were obtained from animals slaughtered at the local slaughter house. The sacrificed animals were on average 1817 days old (range: 515-3791 days). The bone were prepared for analytical studies in the fresh state. They were not stored. The organic matter including both meat and marrow was removed from the bones mechanically in boiling water and treatment with 1% H2O2 (degreasing). The bones were cut into smaller pieces using a saw. The bone were not dried, rather the organic matter including both meat and marrow was removed from the bones mechanically in boiling water and treatment with 1% H2O2 (degreasing). This information was added to the Methods section.

Did the procedure involve the entire bones from the bovine samples? Given the size of bovine bones, this seems unlikely. Please specify which part of the bone was used for analysis and whether the samples were pooled or analyzed individually.

This clarification is crucial, as later in the manuscript you discuss the analysis of trabecular and cortical bone, which differ significantly in terms of bone turnover, mineral content (including all macro- and micro-minerals, not just calcium and phosphorus), and bulk density. Thus, this clarification is especially important chemical analysis studies ICP-OES and ICP-MS.

The midshaft region of the tibial bone was examined. The samples were analyzed individually. This information is now added to the Methods section.

Methods: Providing the explicit formulas used in the calculations would significantly enhance the value of this work. If the authors' intention is to create a reference work showcasing the advantages of 'geophysical' methods in bone analysis, I strongly recommend including a detailed description of how the specific parameters measured by XRD and TA techniques were calculated, similar to what was done for FTIR. Rather than simply stating 'according to standard procedures,' offering clear access to all necessary formulas and mathematical models would ensure that other researchers can accurately replicate these bone analyses in their future work.

As requested, in both cases the “according to standard procedures” statement is now replaced by more detailed description of the XRD and TGA methods.

Qct: Please provide precise information regarding the specific areas of the bone that were analyzed—where exactly was the cortical bone assessed (e.g., epiphysis, diaphysis, or metaphysis, and in which specific regions), and which region was chosen for the trabecular bone (e.g., distal or proximal epiphysis, metaphysis, or diaphysis)? Additionally, it should be clearly stated in the introduction that this study correlates the results of 'geophysical' methods with Peripheral Quantitative Computed Tomography (pQCT), not conventional QCT. This distinction is important, as pQCT, with a significantly lower resolution (0.59 mm voxel size in the present study), differs from the high resolution of standard QCT (potentially up to 0.01 mm), which limits the ability to accurately determine parameters such as trabecular thickness. In turn, the advantage of pQCT is its applicability in in vivo studies on the human forearm, as it is a rapid and non-invasive technique.

Many thanks. This information is now added to the QCT part of the Methods section. We clarified that it was a peripheral QCT and the regions. Epiphysis was assessed for trabecular and diaphysis for cortical bone.

The Methods section lacks a 'Statistical Analysis' subsection, which is essential since correlation, and univariable and multivariable regression analyses appear to be a key components of this work. Please include detailed information on the statistical methods and tests used to evaluate the data.

We are very sorry that this section was missing. Now we added this.

L215: Does 'age' refer to the age of the animals or the duration for which the isolated bones were stored before analysis?

This is the age of the animals, we now clarified this in the text.

L298: When describing correlations (a strictly defined statistical term), avoid using the word 'associations.' These terms are not interchangeable, and 'correlation' should only be used when referring to statistically verified relationships.

Many thanks, we now clarified this, and changed the terms accordingly. In most cases we describe statistically confirmed correlations and not associations.

The conclusions section needs revision, but this can be properly addressed once the precise research hypothesis and the aim of the study are clarified.

Yes, as we now changed the term “geological” throughout the text to geophysical/geochemical, also clarified our scope, we revised the Conclusion section. We now pointed out more strongly why this study was performed, described the collaboration between medical and geophysical groups and also pointed out the novelties of this study.

Round 2

Reviewer 1 Report

Comments and Suggestions for Authors

All the comments have been addressed although it could have been great to nuanced the voxel size given by the instrument in regards to the BMD. 

Author Response

Many thanks, at this point we could not add more about the role of voxel size. We used the very same instrument before in two other human RA studies. Many thanks for the very positive reply. 

Reviewer 2 Report

Comments and Suggestions for Authors

Thank you for the corrections made. I have only one remaining issue with the sentences on L369-370. To improve clarity, I suggest revising the sentence as follows (or simillar):

"The c-axis parameter represents the length of the c-axis of apatite. Using standard Bragg's Law, the interplanar spacing d002 was calculated from the diffraction angle of the (002) reflection. The c-axis length (in Å) was then determined by multiplying by 2."

Author Response

Many thanks for the positive reply. We now change that sentence as you recommended. Many thanks again